# Microglia and Other Cellular Mediators of Immunological Dysfunction in Schizophrenia: A Narrative Synthesis of Clinical Findings

**DOI:** 10.3390/cells12162099

**Published:** 2023-08-19

**Authors:** Khoa D. Nguyen, Andrea Amerio, Andrea Aguglia, Luca Magnani, Alberto Parise, Benedetta Conio, Gianluca Serafini, Mario Amore, Alessandra Costanza

**Affiliations:** 1Department of Microbiology and Immunology, Stanford University, Palo Alto, CA 94305, USA; khoa.d.nguyen@gmail.com; 2Tranquis Therapeutics, Palo Alto, CA 94065, USA; 3Section of Psychiatry, Department of Neuroscience, Rehabilitation, Ophthalmology, Genetics, Maternal and Child Health, University of Genoa, 16126 Genoa, Italy; andrea.amerio@unige.it (A.A.); andrea.aguglia@unige.it (A.A.); benedetta.conio@sanmartino.it (B.C.); gianluca.serafini@unige.it (G.S.); mario.amore@unige.it (M.A.); 4IRCCS Ospedale Policlinico San Martino, 16132 Genoa, Italy; 5Department of Psychiatry, San Maurizio Hospital of Bolzano, 39100 Bolzano, Italy; magnani1991@gmail.com; 6Geriatric-Rehabilitation Department, University Hospital of Parma, 43126 Parma, Italy; aparise@ao.pr.it; 7Department of Psychiatry, Adult Psychiatry Service, University Hospitals of Geneva (HUG), 1207 Geneva, Switzerland; 8Department of Psychiatry, Faculty of Biomedical Sciences, University of Italian Switzerland (USI), 6900 Lugano, Switzerland; 9Department of Psychiatry, Faculty of Medicine, University of Geneva (UNIGE), 1211 Geneva, Switzerland

**Keywords:** schizophrenia, microglia, neuroinflammation, immunological dysfunction, clinical correlates

## Abstract

Schizophrenia is a complex psychiatric condition that may involve immune system dysregulation. Since most putative disease mechanisms in schizophrenia have been derived from genetic association studies and fluid-based molecular analyses, this review aims to summarize the emerging evidence on clinical correlates to immune system dysfunction in this psychiatric disorder. We conclude this review by attempting to develop a unifying hypothesis regarding the relative contributions of microglia and various immune cell populations to the development of schizophrenia. This may provide important translational insights that can become useful for addressing the multifaceted clinical presentation of schizophrenia.

## 1. Introduction

Schizophrenia (SCZ), a chronic psychiatric illness that affects approximately 24 million people worldwide, is characterized by the hallmark “positive” symptoms of hallucinations and delusions and “negative” symptoms of apathy, anhedonia, avolition, and emotional and cognitive impoverishment [1]. This debilitating disorder imposes a significant risk of physical and mental health complications, ranging from coronary heart disease to suicidal behavior (SB), highlighting reciprocal relationships between somatic psychic implications in neuro-psychiatric conditions [2]. SCZ diagnosis is difficult due to the spectral nature of the illness and the complex progression of its clinical manifestation. Patients with SCZ often present with subtle irritation/behavioral changes in the prodromal phase, followed by the onset of psychosis. Before 2013, SCZ was categorized into various subtypes (paranoid, disorganized, catatonic, undifferentiated, and residual) based on specific clinical presentations. However, this discrete division of the illness was supplemented by the concept of SCZ being a spectral disease that includes schizoaffective, schizophreniform, and schizotypal personality disorders [3].

Various environmental and genetic risk factors have been implicated in the etiology of SCZ, with exposure to toxins/infectious agents, urban lifestyle, pregnancy complications, substance abuse, family history, and male sex reportedly associated with an increase in SCZ risk [4]. However, despite the discovery of many SCZ-associated genetic variants, to date, no single gene has been identified as the dominant causative factor of SCZ development. Currently, there are no effective treatments of SCZ as antipsychotics are only capable of suppressing positive symptoms [5]. Therefore, further research into the cellular and molecular pathology of SCZ is integral to the therapeutic development for this psychiatric disorder.

Consistent with the neuropsychiatric nature of SCZ, various central nervous system (CNS) abnormalities of affected individuals have been observed, including enlarged ventricles, reduced gray matter volume, smaller hippocampus, decreased brain asymmetry, and neurochemical disturbances [6]. Concurrently, accumulating evidence from neuroimmunological studies is pointing to the immune system’s potential involvement in SCZ development, an analogy to other major psychiatric conditions [7,8]. While several prominent hypotheses, including immunocytokine-driven inflammation [9], innate immune-mediated dysfunction in synaptic pruning [10], and antibody-mediated autoimmunity [11], have been suggested, these hypotheses were predominantly derived by extrapolating SCZ-like animal models and human genetic association studies/fluid-based molecular biomarker analyses. Since evidence on cellular abnormalities in SCZ remains scarce and somewhat scattered in the literature, this review aims to provide a comprehensive distillation of clinical findings regarding the potential involvement of various immune cell types in SCZ to facilitate the development of a unifying hypothesis of immune dysregulation in this psychiatric illness.

## 2. Cellular Constituents of CNS Immunological Aberrations in SCZ

Following reports of elevated neuroinflammation in SCZ, an initial hypothesis stated that CNS immune disturbance may be involved in SCZ pathogenesis. The subsequent discovery of risk factors in genes associated with immune-mediated neurodevelopmental processes and the identity of CNS immune cells provided further support for the immunological origin of SCZ (Figure 1). This section highlights important evidence of CNS immune-related cell-based changes in SCZ and provides a discussion of potential caveats in the interpretation of these findings.

### 2.1. Microglia

Microglia are the resident brain innate immune cells that have been implicated in host defense against neurotropic pathogens, brain development, and neurodegenerative disorders [12]. The growing importance of these cells in behavioral illnesses is also highlighted by the growing attention they have received in neuroimmunological investigations studying possible alterations in their distribution and function in SCZ. Given the difficulty of sampling live human microglia, cell characterization in SCZ has been mostly conducted in post mortem brain samples. One of the earliest studies of microglia in SCZ was an analysis of embryonic microglia derived from female patients with SCZ in whom these cells displayed a highly phagocytic phenotype compared to healthy controls (HCs) with no psychiatric illnesses [13]. Subsequently, morphologically activated microglia have been observed in the prefrontal cortex (PFC) and visual cortex of paranoid and chronic patients with SCZ in close proximity to dystrophic oligodendrocytes [14,15,16]. Further subcategorization of patients with SCZ revealed that this abnormal microglia activation phenotype might contribute to oligodendrocyte dystrophy in schizophrenia patients with positive symptoms [17,18], providing some of the first morphological evidence for the possible involvement of microglial activation in the development of these SCZ-associated pathologies.

In addition to these morphometric studies, others attempted to localize activation markers on microglia by immunohistochemical analysis and found increases in HLA-DR+ activated microglia in the frontal/temporal cortex and the hippocampus in patients with SCZ [19,20]. Of note, these activated microglia exhibited some degenerating features [21] and were reportedly associated with interleukin IL1β expression in the PFC [22]. Microglia activation was also observed in some of these brain regions in patients with Alzheimer’s disease (AD) and affective disorders [23,24], suggesting the possible existence of microglia reactivity against a common dysfunctional neuronal circuit among various CNS disorders. However, microglia activation in SCZ and affective disorders remains to be validated as some studies failed to detect changes in HLA-DR+ microglia in various brain regions, including PFC, anterior cingulate cortex (ACC), and hippocampus, and/or attribute this microglia activation profile to death by suicide [25,26]. Besides HLA-DR expression, a unique microglial proteome might exist in SCZ. For example, S100 calcium-binding protein (S100) A8/A9 expression [27], an inflammatory marker, was found to be upregulated in frontal cortex microglia, while quinolinic acid expression, a neuroprotective molecule, was suppressed in CA1 hippocampal microglia in patients with SCZ [28]. Furthermore, patterns of expression of the purinergic receptor (P2RY12) were not altered significantly in SCZ microglia compared to HCs, while downregulation of this marker was a cardinal feature of microglia in multiple sclerosis and AD [29].

Microgliosis, marked by increased ionized calcium-binding adapter molecule 1 (IBA-1) staining density [30], has also been linked to the characteristic anatomical lateralization in the ACC of patients with SCZ, as well as patients with bipolar disorder (BD). However, this abnormality was not observed in the PFC of SCZ patients [31], in which microglia showed synaptic pruning activity comparable to HC counterparts. This discrepancy may be attributed to region-specific alterations in microgliosis in SCZ as a significant increase in microgliosis was confirmed in the frontal and temporal cortex, but in no other regions [32]. Alternatively, age-dependent microgliosis may account for the discrepancy between the aforementioned studies [33,34].

To circumvent several methodological shortcomings of post mortem studies, several studies conducted microglia imaging in living patients with SCZ using translocator protein (TSPO) tracer analysis. TSPO is a receptor mainly found on the outer mitochondrial membrane that is expressed throughout the body and brain [35,36]. Among other functions, TSPO can modulate the immune system through modulation of oxidative bursts by neutrophils and macrophages, inhibition of the proliferation of lymphoid cells, and secretion of cytokines by macrophages [37]. Expression of TSPO has also been linked to inflammatory responses that occur after vascular brain injury and in some neurodegenerative or mixed neurodegenerative/vascular diseases, in which interesting links with neuropsychiatric disorders are currently being investigated [24,38]. There are significant inconsistencies among studies using TSPO tracer analysis to investigate possible alterations in the TSPO signal [39,40,41,42,43,44,45], and even studies conducting systematic analyses of single-patient data could not reach a consensus on changes in TSPO signal in SCZ brains [46,47]. This discrepancy may be due to the lack of specificity of TSPO to microglia, as various cell subsets, including astrocytes and endothelial cells, also showed this signal [48].

To date, cell-based evidence of the involvement of microglia in SCZ pathogenesis remains inconclusive. While we cannot discuss this in detail here, numerous association studies also suggested the presence of an increase in SCZ risk in various genes involved in microglia-mediated neuroinflammation [49] and synaptic pruning [50], as well as microglia homeostasis [51]. Of note, many of these molecules are not exclusively related to microglia function and identity, which highlights the urgent need for more accurate characterizations/confirmation of possible changes in various aspects of microglia in this psychiatric illness. Further studies with induced pluripotent stem-cell-derived microglia-like cells may shed some additional light on the role played by microglia genetics in SCZ. Additionally, the development of a rapid, efficient, and reliable method to isolate microglia from post mortem brain tissues for high-throughput proteomic and transcriptomic analyses is expected to provide additional clarification regarding the involvement of this innate immune cell type in SCZ.

### 2.2. Other Immune Cell Types

Besides microglia, abnormalities in other immune cells have also been detected in CNS samples of SCZ patients(). For example, dynamic trafficking of adaptive immune cells in the CNS has been linked to SCZ. Whole brain immunohistochemical quantitation of T cell and B cell frequencies showed marked increases in these lymphocytes in patients with SCZ and affective disorders compared to HCs [52]. Spatial analyses also revealed region-specific alterations of these lymphocytes in SCZ brain tissues. In this regard, immunohistochemical analysis in the dorsal PFC (DPFC) revealed a reduction in CD3+ T cell density in the leptomeningeal space of subjects with SCZ compared to HCs and no significant difference in the frequencies of these lymphocytes in the gray matter of both groups. Furthermore, analysis of hippocampal T cell (CD3) and B cell (CD20) mRNA transcripts revealed a significant increase in CD20 and CD3 expression in residual SCZ (characterized by negative symptoms) compared to paranoid SCZ (characterized by positive symptoms) and HCs, suggesting that adaptive immune cell trafficking to the hippocampus might be associated with negative symptoms of SCZ [19].

Monophagocyte-related alterations have also been reported in SCZ brains. In the neurogenic subependymal zone (SEZ), an SCZ subgroup with high inflammation (HC) (defined by elevated expression of IL-1β, IL-1R1, serine protease inhibitor member 3 [SERPINA3], and c-x-c motif chemokine ligand 8 [CXCL8] mRNA transcripts) showed higher expression of the identity markers of macrophages (CD163) and monocytes (CD14) than high-inflammation HCs [53]. Notably, increased infiltration of monophagocytes into the SEZ appeared to be a shared pathological feature between patients with SCZ and patients with BD [54]. In the mid brain, immunostaining revealed that CD163+ macrophage density was elevated in high-inflammation SCZ compared to HCs [55]. Along with the close association between these cells and dopaminergic neurons in the substantia nigra, a positive correlation between CD163 and the complement C1 subcomponent A (C1qA) mRNA transcripts was detected in high-inflammation SCZ, suggesting a possible involvement of dysregulated complement-mediated phagocytic activity of these cells in the development of inflammatory pathology in SCZ.

Nevertheless, the exact role played by these brain immune cells in SCZ remains contentious. For instance, immunohistochemical analysis in the DPFC of SCZ patients yielded no evidence of CD163+CD206+ perivascular macrophage infiltration into the brain parenchyma [33]. In contrast, a different study that transcriptionally quantified CD163 mRNA expression in the DPFC showed that macrophage accumulation in this brain region was a signature of high-inflammation SCZ [56]. These discrepancies might be attributed to different analytical approaches employed by these studies (mRNA vs. protein expression), as well as the heterogeneity of the SCZ cohort (i.e., the presence of high inflammation or acute psychosis). Alternatively, procedural differences in tissue collection and storage might also affect the detection of the macrophage marker of interest.

Besides post mortem brain studies, cerebrospinal fluid (CSF) analysis of adaptive and innate immune cells in SCZ is another approach that has been investigated. For example, acute psychotic symptomatology in SCZ patients [57] was associated with an accumulation of monophagocytes in CSF samples. Further analysis revealed that this signature of innate immune alteration was accompanied by an increase in the frequency of lymphocytes with an activated phenotype during psychosis onset in SCZ [58]. Interestingly, this abnormality was responsive to conventional neuroleptic medication, which led to post-treatment normalization of monophagocyte counts in several subjects with SCZ. Alterations in CD4+ and CD8+ T cells have also been reported in acutely psychotic SCZ [59]. Lastly, B cells in the CSF of SCZ patients showed marked differences in their antibody repertoire compared to HCs [60], hinting at a potential involvement of distinct pathogenic B cell subsets in autoimmune-like symptoms of SCZ.

Collectively, these aforementioned studies highlighted the potential involvement of CNS adaptive and innate immune cells in distinct SCZ-associated pathologies. While excessive neuroinflammation may be uniquely linked to changes in monophagocytes, negative symptoms of SCZ may be related to alterations in adaptive lymphocytes. Importantly, perturbations in both cell types might be associated with the onset of psychosis and have been responsive to antipsychotic treatments. Some of these CNS immunological disturbances could also represent an overlapping pathological feature among different neuropsychiatric illnesses.

## 3. Alterations in Circulating Immune Cells in SCZ

While most studies of CNS immunity have focused on microglia, some also attempted a more diverse characterization of the potential contribution of different peripheral immune cell subsets in SCZ (Table 1). Findings on changes in innate immune cells, including monocytes and neutrophils, along with those related to autoantibodies, presumably produced by specific pathogenic B cell clones, appear to be the most consistent and/or have the lowest risks of bias [61], thus supporting the inflammatory and autoimmune hypotheses of SCZ pathophysiology. Various reports also suggested that natural killer (NK) cells and different T cell populations in SCZ were dysregulated. In this context, this section summarizes the major findings on peripheral immunological changes, including the proposed clinical implications and associated mechanistic insights.

### 3.1. Monocytes

In the innate immune system, monocytes represent the counterparts in the circulatory system of microglia and are known for their plasticity in responding to environmental changes. There is a growing body of clinical evidence of monocyte alterations in SCZ blood samples [62] (Figure 2), including higher total monocyte counts in SCZ during the first episode of psychosis, although some discrepancies remain as to whether these alterations are linked to disease severity [63,64]. A similar increase in total monocyte number was also observed in patients with non-affective psychosis [65], while elevated counts of classical monocytes and proinflammatory monocytes have been linked to clozapine-treated and recent-onset SCZ, respectively [66,67]. Several studies found higher monocyte-related indices in SCZ. For example, a large study involving over 6000 patients with SCZ revealed that the monocyte to lymphocyte ratio (MLR) was significantly increased in SCZ compared to HCs [68]. Notably, this ratio could be used to distinguish SCZ (during the first episode of psychosis) from HCs or patients during the first episode of depression [69]. Furthermore, while elevated MLR may represent a shared pathological hallmark of SCZ and BD compared to HCs, some discrepancies remain, possibly related to methodological variations and differing patient inclusion criteria [70,71]. Similar relationships were observed between monocyte counts and other metabolic markers, such as the cardioprotective HDL (monocyte to HDL ratio, MHR), hinting at a possible involvement of these cells with the onset of cardiometabolic comorbidities in SCZ and BD [72,73,74,75].

These observations of alterations in various monocyte features represent a significant breakthrough in SCZ research. While many studies inferred a monocyte-associated gene set from bulk immune cell transcriptome profiling, the studies mentioned above typically focused on a phenotypic and functional characterization of the monocytes themselves. For example, an interferon gene signature in isolated monocytes was observed in SCZ, with dynamic changes over the disease course [77]. Interestingly, unique alterations in protein tyrosine phosphatase non-receptor type 7 (PTPN7)/NGFI-A-binding protein 2 (NAB2) were observed between SCZ and BD, while some overlapping gene signatures, characterized by elevations in activating transcription factor 3 (ATF3)/dual specificity phosphatase 2 (DUSP2) and early-growth-response protein 3 (EGR3)/mad-max dimerization protein (MXD1) [78], were shared between both illnesses. Regarding the activation phenotype, a higher expression of triggering receptor expressed on myeloid cell (TREM) 1 and 2 was documented in monocytes from patients with SCZ, with the former having been linked to transcriptional changes in ATF3 and EGR3 [79,80]. Importantly, these changes were specific to SCZ but not patients with major depressive disorder (MDD), thus providing further evidence of the utility of monocyte-related markers as a distinguishing feature between these two neuropsychiatric conditions. Increased expression of a canonical activation marker, HLA-DR, was also observed in SCZ monocytes, along with changes in their phagocytic activity during acute psychotic onset [81,82]. Alterations in cytokine production were also observed in inflammatory mediators, such as the production of IL-1, IL-6, and tumor necrosis factor (TNF-α) [83] from SCZ monocytes, which was accompanied by a higher response to lipopolysaccharide (LPS)/toll-like receptor 4 (TLR4) stimulation [84]. Additionally, concanavalin (Con-A) stimulated IL1 secretion from monocytes in peripheral mononuclear cells from drug-naïve SCZ patients was elevated [85], while polyinosinic-polycytylic acid (poli I:C)/toll-like receptor 3 (TLR3)-stimulated intracellular production of IL1 from monocytes was reduced compared to HCs [86]. Other changes in baseline TLR4 expression and TLR4 downregulation in response to LPS were also observed in monocytes from SCZ with tardive dyskinesia [87] and in those from patients with first episode of psychosis [88].

Of potential clinical utility is the presence of various monocyte-specific markers for treatment response monitoring and differential diagnosis of SCZ. Specifically, reduced glucose transporter (GLUT1) expression in monocytes has been proposed as a key diagnostic feature to distinguish SCZ from BD, MDD, and autistic spectrum disorder [89], while soluble CD14, an identity marker of circulating monocytes, could accurately predict subsequent SCZ diagnosis [90]. A monocytic transcription signature was also proposed as a candidate marker for monitoring beneficial simvastatin response in patients with SCZ [91]. The effectiveness of other antipsychotics, such as haloperidol/perazin and clozapine, can also be predicted by a reduced monocyte production of I-1/TNF-α and reactive oxygen species (ROS), respectively, while the effectiveness of olanzapine could be monitored by pre-treatment monocytic expression of the fatty acid receptor CD36 [83,92,93]. Altogether, these findings on distributional, phenotypic, and functional alterations in monocytes are of high clinical relevance to provide an improved understanding of the cellular mechanisms of SCZ initiation/progression, as well as the regression of its clinical symptoms by currently available pharmacologic agents.

### 3.2. Granulocytes

Circulating granulocytes consist of three major myeloid cell subsets, namely basophils, eosinophils, and neutrophils. While the first two are rarely discussed in the context of neuropsychiatric illnesses, an increase in neutrophil-related parameters represents one of the most consistent findings regarding changes in peripheral immune cells in SCZ (Figure 2). Several studies involving over 6000 patients found a marked increase in the total number of circulating neutrophils during acute psychotic symptomatology in SCZ patients [68] and in patients with a first episode of psychosis [94,95,96]. Notably, in the latter, the increase in neutrophil count was associated with various CNS anatomical pathologies often observed in SCZ [97], including enlarged ventricles and reduced gray matter volumes, as well as hallmark symptoms of this illnesses, such as hallucination and avolition. Comparable increases in neutrophil count were also noted in other subtypes of SCZ, including paranoid, residual, and non-affective psychotic patients [98,99]. Similarly, the neutrophil to lymphocyte ratio (NLR), an index commonly used for clinical assessment of inflammation, was higher in SCZ patients, with some sex-specific differences [100,101], and in patients affected by MDD and BD [102,103,104,105]. Of note, NLR might represent a common feature of immunological disturbance among various types of neuropsychiatric illnesses, ranging from bipolar disorder and major depression to SCZ, although it remains unclear how the actual magnitudes in NLR change should be interpreted to differentiate SCZ from manic episodes of BD [70,106]. Furthermore, despite the consensus on increased NLR in SCZ, conflicting findings exist regarding its potential association with various symptoms and disease stages of SCZ. For instance, while increased NLR yielded no correlation with SCZ severity and symptoms as assessed by the Brief Psychiatric Rating Scale (BPRS) [107], the correlation became significant once different scales were used, e.g., the Positive and Negative Syndrome Scale (PANSS) [64,108], the Clinical Global Impression-Severity Scale (CGI-S) [109], and the Brief Negative Symptom Scale [100,110]. More specifically, a recent large-scale analysis based on BPRS and CGI-S involving over 1000 SCZ patients demonstrated a significant association between symptoms and NLR. Longitudinal changes in NLR between disease relapse (increased NLR) and remission (reduced NLR) have also been reported [111,112,113]; however, a large-scale study involving 618 patients found no differences between these two disease stages [114]. Collectively, these conflicting results might be attributed to the complexity of the progression of clinical symptoms in SCZ, which is often compounded by the impact of various demographic factors (age/sex) and medication status on neutrophil numbers. In fact, the agranulocytosis effect of clozapine [115] might account for the observed reduction in NLR after treatment in SCZ [109,116], indicating the potential clinical utility of this immunological marker for efficacy monitoring of this atypical antipsychotics.

Besides changes in cell abundance, studies also reported various functional and phenotypic alterations of neutrophils in SCZ [116,117,118,119,120,121], including oxidative stress [108], which showed a positive association with NLR counts in SCZ. Consistent with this finding, some authors observed that SCZ neutrophils exhibited an increased expression of various markers of oxidative stress, such as malonaldehyde [122] and superoxide anion [123,124]. While several studies reported increased phagocytic activity of SCZ neutrophils, the magnitude of the observed increase differed significantly [81,124]. In summary, while certain details regarding the mechanistic involvement of neutrophils in various clinical aspects of SCZ pathogenesis remain to be confirmed in large-scale studies, most of the evidence suggests that these sentinel innate immune cells play an important role in peripheral immune dysfunction/inflammation in SCZ.

### 3.3. Natural Killer Cells

NK cells are a type of immune cell with both adaptive and innate features. They have been implicated in a wide range of human diseases, ranging from infection and cancer to CNS disorders. Potential abnormalities in both NK cell count and function have been documented in SCZ, although with significant discrepancies among studies. Flow cytometric analysis showed increased counts of NK cells in clozapine-treated chronic SCZ blood samples compared to HCs [67]. In contrast, computational deconvolution based on gene expression yielded lower NK cell numbers in both drug-naïve and medicated SCZ patients [125,126], and this decrease was uncorrelated to psychotic relapse/remission. A different flow cytometric study confirmed these lower NK cell counts in chronic SCZ; however, medication appeared to increase NK cell numbers [127]. These differences could be due to the different quantification methods used (flow cytometry vs. gene expression, whole blood vs. peripheral mononuclear cells). Chronic SCZ subpopulations who received differing regimens of antipsychotic drugs may exhibit distinct NK cell profiles. In fact, an earlier study involving a heterogenous SCZ cohort (more than four subtypes and treatment modalities) failed to detect any abnormalities in immune cell counts in blood samples [128]. Regarding the function of NK cells in SCZ, there have been conflicting reports on NK cell cytolytic activity, possibly related to significant variations in general NK cell lytic function, medication regimes, and SCZ subtypes [129,130,131,132]. With regard to phenotypes, some studies suggested that an elevated expression of NK-cell-activation markers, such as HLA-DR and natural killer group 2C (NKG2C), might be associated with the first episode of psychosis in SCZ patients compared to HCs. However, these features were also observed during the first psychosis of patients with BD [133]. Another inflammatory marker of blood–brain barrier (BBB) disruption, S100B, was reportedly elevated in NK cells of SCZ patients during acute psychotic symptomatology compared to HCs, possibly associated with the activation of stress signaling pathways [134]. In contrast, in medicated SCZ patients, disease remission was linked to a higher NK cell production of IL17 [135]. Given these numerous discrepancies, the involvement of NK cells in SCZ requires further validation studies with a comprehensive analysis of all parameters of NK cell phenotype and function among well-categorized SCZ and appropriately matched HC cohorts.

### 3.4. B Lymphocytes

Considering that various autoantibody types have been found to be elevated in serum and CSF samples of SCZ patients, B cells, as producers of antibodies, have long been implicated in the autoimmune hypothesis of SCZ (Figure 2). For instance, several small-cohort studies suggested that autoantibodies may act against anti-glutamic acid decarboxylase (GAD), γ-aminobutyric acid A receptor 1 (GABAR1), anti-acetylcholine receptor (A7ChR), and N-metil-D-aspartato receptor (NMDAR) in SCZ pathogenesis [136,137,138,139]. However, large-scale studies (*n* > 150) failed to confirm the biological significance of these autoantibodies in this disorder [140,141,142], or they detected only a small range of autoantibodies in low concentrations in certain SCZ subsets, such as clinically high-risk SCZ, SCZ with the first episode of psychosis, and SCZ with tardive dyskinesia. Another set of small-cohort studies found a high proportion (>20%) of autoantibodies against various classes of molecules, including DNA/RNA species [143,144], heat-shock proteins [145,146], and neuronal/neurotransmitter targets [147,148,149,150], which currently await further independent validation with larger sample sizes. Finally, several large-scale studies reported the elevated expression of circulating antibodies against IL-1, IL-6, IL-8, CD25, and gliadin (which has also been linked to peripheral inflammation) in SCZ patients, suggesting a possible mechanistic involvement of dysregulated inflammation in the B-cell-driven antibody-mediated autoimmune hypothesis of SCZ [151,152,153,154,155].

Besides autoantibody characterization, only few studies examined potential alterations in circulating B cell subsets in SCZ. Of note, circulating CD19+ B cells were elevated in paranoid SCZ patients during acute psychotic symptomatology, which could be suppressed by treatment [156]. Blood samples of clozapine-treated patients with chronic SCZ showed elevated levels of naive IgD+CD27-CD19+ B cells [67]. CD5+ B cell counts were also elevated in patients with SCZ; however, this increase remained unaffected by antipsychotic withdrawal [157,158]. Furthermore, the association between CD5+ B cells and SCZ was disputed by a study that considered the confounding effect of race, finding that African Americans, regardless of their disease status (SCZ or HC), appeared to have higher levels of CD5+ B cells than Caucasians [159]. Taken together, these findings warrant further multidimensional analyses of B cell repertoire and function in SCZ, which is pivotal to elucidate the mechanistic underpinnings of the various autoimmune-like pathologies in selected SCZ subgroups.

### 3.5. T Lymphocytes

Numerous phenotypic studies of various circulating T cell subsets have been conducted in SCZ (Figure 2). While the findings were inconsistent with regard to the distribution of total CD3+ T cells, CD4+ helper T cells, and CD8+ cytotoxic T cells [127,156,160,161,162], they mostly agreed with regard to increased T cell activation [66,163,164]. Furthermore, activated T cells in medicated SCZ patients appeared to have higher levels of CD25 than drug-naïve patients. Along with this ex vivo-activated phenotype, several studies observed a reduced responsiveness to in vitro stimulation of IL-2 production in T cells from drug-naïve SCZ patients compared to HCs [165,166,167]. However, this decrease in IL-2 production was not observed in a study of paranoid and residual SCZ [99], possibly due to how patients were stratified in this study and/or the use of purified T cells/peripheral blood mononuclear cells vs. whole blood for in vitro assays. Regarding T-helper (Th) cell subtypes, an increase in Th2 cells was reportedly associated with the SCZ subtype with a pro-inflammatory monocyte feature [66]; however, this alteration could not be confirmed by a different study [99]. The same two studies also yielded inconsistent findings regarding the role of Th1 cells. In contrast, consensus exists regarding the elevated numbers of regulatory T cells (Treg) and IL-17-producing Th cells (Th17) in SCZ [66,135,163,168,169,170]. Of note, the Treg increase in medicated SCZ patients was associated with fewer negative symptoms [169], while Th17 elevation might be linked to psychosis [168,171]. Interestingly, the Th17 increase in SCZ with the first episode of psychosis was suppressed by risperidone treatment [168], while haloperidol or risperidone-treated SCZ patients continued to show elevated Th17 numbers during stable remission [135]. These paradoxical findings warrant further investigations into the longitudinal effects of antipsychotics on Th17 cell counts in SCZ. Mucosal-associated invariant T (MAIT) cells [170] were also elevated in patients with SCZ, supporting the hypothesis of mucosal microbiome involvement in SCZ pathogenesis.

Several studies investigating the T cell phenotype examined the expression of dopamine receptors, the presumed target of clozapine [172]. However, their findings on the expression of dopamine receptors D (DRD) 2 and 4 were inconsistent, likely due to methodological differences (mRNA expression vs. flow cytometry) [173,174]. T-cell-specific oxidative damage in SCZ has also been quantified and showed an increase in mitochondrial dysfunction across different T cell subsets from patients with acute relapse compared to HCs [175]. Furthermore, this cellular pathology appears to be associated with positive symptoms [175]. Lastly, T cell methylation profile and repertoire were reportedly associated with distinct SCZ subtypes. Widespread methylation markers in T cells were observed in SCZ patients with more severe symptoms and cognitive impairment [176]. Additionally, some cis-diagnostic (SCZ specific) and trans-diagnostic (common among several psychiatric illnesses) genetic variants also appeared to be epigenetically active in CD4+ T cells, but not innate immune cells, from SCZ patients, indicating the potential utility of these T-cell-specific biomarkers for individualized therapeutic development in SCZ [177].

Overall, studies examining the role of T cells in SCZ point to possible alterations in the activation status of this lymphocyte subset in SCZ, typically consistent with an increase in the expression of the canonical activation marker CD25 and elevated counts of Treg and Th17. To obtain a more comprehensive understanding of the role of T cells in SCZ, future studies should attempt more detailed characterizations of other T cell subsets, as well as different immunometabolic markers of T cells across the entire spectrum of SCZ patients with different symptoms and treatment statuses.
cells-12-02099-t001_Table 1Table 1Major immunological dysfunctions in patients with schizophrenia (SCZ).Central Nervous SystemFeaturesAnatomical LocationMicrogliaIncreased phagocytosis [13]Whole brain
No change in phagocytosis [31]PFC
Increased IBA-1 [30]ACC
No change in IBA-1 [31]PFC
Increased HLA-DR [19,20]Frontal/temporal cortex, hippocampus
No change in HLA-DR [25,26]PFC, ACC, hippocampus
Increased S100A8/9 [28]Frontal cortex
Activated morphology [14,15,16]PFC, visual cortex
Microgliosis [32]Frontal/temporal cortex
Decreased quinolinic acid [29]HippocampusT cellsIncreased CD3+ frequency [19,48]Whole brain, hippocampus
Decreased CD3+ frequency [19]Dorsal PFC
Increased CD4+ frequency [55]CSF
Increased CD8+ frequency [55]CSFB cellsIncreased CD20+ frequency [19,52]Whole brain, hippocampus
BCR alterations [60]CSFMacrophagesIncreased CD163 [53,55,56]Dorsal PFC, CSF, SEZ
No change in CD163 [33]Dorsal PFC
Increased CD14 [53]SEZ
Increased frequency [57]CSF**Peripheral Blood****Features**MonocytesIncreased total monocyte counts [62,63,64]
Increased classical monocyte counts [66]
Increased pro-inflammatory monocyte counts [67]
Increased monocyte to lymphocyte ratio [68]
Increased monocyte to HDL ratio [72,73]
Increased TREM1/2 [79,80]
Increased ATF3/EGR3 [78]
Increased HLA-DR [82]
Increased phagocytosis [81]
Increased IL-1, IL-6, TNF-α [83,85,86]
Increased CD36 [93]
Increased reactive oxygen species [92]
Alterations in TLR4 signaling [84,87,88]
Unique interferon gene signature [77]
Reduced GLUT1 [89]GranulocytesIncreased neutrophil counts [68,94,95,96,97,98,99]
Increased neutrophil to lymphocyte ratio [100,101]
Increased oxidative stress [108]
Increased malonaldehyde [122]
Increased superoxide anion [123,123]
Increased phagocytosis [81,124]Natural killer cellsIncreased total counts [68]
Decreased total counts [125,126,127]
No change in total counts [128]
Increased cytotoxicity [132]
Decreased cytotoxicity [129,130]
No change in cytotoxicity [121]
Increased NKG2C [133]
Increased S100B [134]T CellsIncreased CD3+ and CD4+ T cell counts [153]
Reduced CD3+ T cell counts [150]
Reduced CD4+ T cell counts [146,152]
Increased activation [63,154,155]
Increased CD25 [156]
Reduced IL2 production [156,157,158]
No change in IL2 production [92]
Increased Treg [63,155,159,160,161,162]
Increased Th17 [63,154,155,156,157,158,159,160,161,162]
Increased mucosal-associated invariant T cells [162]
Increased oxidative stress [166]
Altered methylation profile [167]
Altered TCR repertoire [167]B CellsIncreased CD19+ B cell frequency [156]
Increased IgD+CD27-CD19+ naïve B cell frequency [67]
Increased CD5+ B cell frequency [157,158]
No change in CD5+ B cell frequency [159]
Increased GAD, GABAR1, A7ChR, NMDAR autoantibodies [136,137,138,139]
No changes in GAD, GABAR1, A7ChR, NMDAR autoantibodies [140,141,142]
Increased DNA/RNA autoantibodies [143,144]
Increased heat-shock protein autoantibodies [145,146]
Increased neuronal marker autoantibodies [147,148,149,150]
Increased IL-1/IL-6/IL-8/gliadin autoantibodies [151,152,153,154,155]


## 4. Implications for Mechanistic Studies and Therapeutic Development

To explore disease mechanisms of SCZ, various in vitro and animal studies have been established, with induced pluripotent stem cells derived from SCZ patients and rodent strains based on human genetics as the most clinically relevant attempts to model SCZ-associated pathology [178,179]. However, most immune-related studies to date have focused exclusively on the role of microglia, but not other peripheral immune cell subsets, in this psychiatric illness. For example, microglia from mice with an overexpression of C4, a genetic risk factor of SCZ, exhibited increased synaptic pruning [180]. Similarly, microglia derived from SCZ patient-derived induced pluripotent stem cells were reportedly more activated and might cause neuronal metabolic disruption [181,182,183]. Building upon these preclinical findings, our synthesis of clinical evidence pertaining to various alterations of other immune cells in peripheral tissues, as well as the CNS, will provide an important impetus for further mechanistic exploration of a potential dynamic interplay among these cell types in different tissue landscapes during SCZ development.

Anomalies in various immune cell subsets and their trafficking patterns in SCZ also suggest the potential utility of cell-type-specific immunotherapy as a novel pharmacological approach for this psychiatric illness [184,185]. In this regard, immunomodulators aiming to inhibit lymphocyte trafficking (fingolimod) and deplete B cells (anti-CD20 monoclonal antibody, rituximab) could significantly improve negative symptoms and general psychopathology of SCZ, respectively [186,187]. Furthermore, cytokine-based immunotherapies, including those aiming to inhibit IL-6 (tocilizumab) and TNF-α (adalimumab), showed promising efficacy in some randomized controlled trials of SCZ [188,189]. Other immunosuppressants, such as azathioprine [190], prednisolone [191], and methotrexate [192], also exhibited preliminary clinical benefits in SCZ patients.

Besides these classical immunotherapies, the potential clinical efficacy in SCZ of other anti-inflammatory treatments, including aspirin, minocycline, N-acetylcysteine, estrogens, telmisartan (an angiotensin receptor 1 antagonist), and pioglitazone (a PPAR-γ antagonist) [193,194,195], has been observed. Interestingly, other anti-inflammatory medications, such as celecoxib, davunetide, dextromethorphan, fatty acids, pregnenolone, statins, and varenicline, did not have a significant impact on SCZ symptoms in a recent meta-analysis, highlighting the presence of inflammatory-pathway-specific abnormalities in SCZ development. Collectively, these findings support the emerging importance of immune-related dysfunction as a targetable pathology of SCZ [196]. However, additional trials are required to confirm these findings as the effectiveness of selected therapies, including anti-IL-6 and anti-CD20 antibodies as well as prednisolone, has been questioned in some clinical studies [197,198,199].

## 5. Concluding Remarks

Our narrative synthesis of the literature on microglia and other cellular mediators of immunological dysfunction in schizophrenia points to the existence of several immunopathological hallmarks of SCZ in circulation, including major distributional abnormalities in neutrophils, monocytes, immunoregulatory T cell populations, and autoantibody repertoire. In addition, there may be evidence of the occurrence of a dynamic trafficking pattern of both adaptive (T and B cells) and innate (microglia and macrophages) immune cells in various anatomical brain regions during different disease stages of SCZ. We also noted numerous inconsistencies in the clinical evidence, which not only reflect the heterogenous nature of this spectral disorder and other demographic and clinical confounders, but also emphasize the urgent need for the development of high-throughput and high-resolution methodological approaches to facilitate more comprehensive investigations of potential abnormalities in the immune system’s cellular compartments. In this regard, cell-based models of SCZ are of high mechanistic utility as the behavioral nature of this illness is likely difficult to be faithfully recapitulated with animal modelling. Furthermore, such cell-based models would yield a clearer delineation of the roles played by genetic vs. environmental factors in SCZ pathogenesis and provide versatile platforms for therapeutic development to address specific unmet needs in the clinical management of SCZ.

## Figures and Tables

**Figure 1 cells-12-02099-f001:**
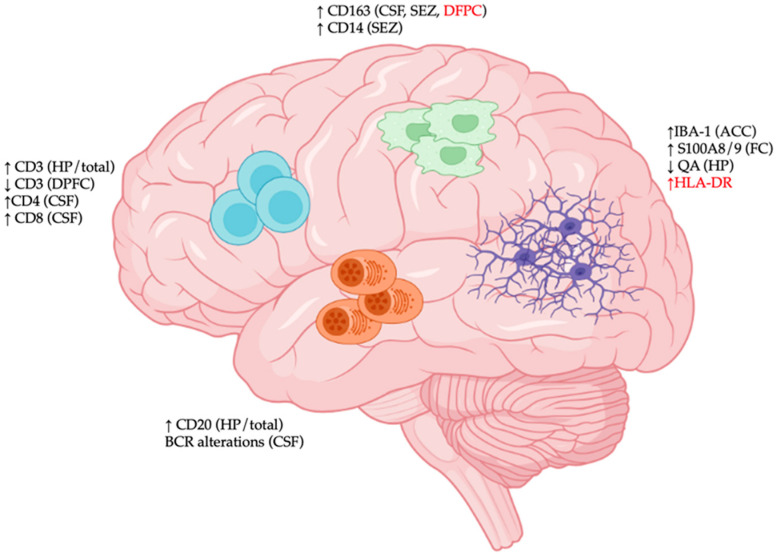
Immunological disturbances in the central nervous system (CNS) of patients with schizophrenia (SCZ). Region-specific immunological changes in CNS tissues of SCZ patients are characterized by (1) elevated expression of various activation markers of microglia (S100/A8, HLA-DR), decreased expression of neuroprotective quinolinic acid (QA), and increased microgliosis (IBA-1 density); (2) increased expression of macrophage markers (CD14 and CD163); (3) dynamic trafficking of various T cell populations (CD3/CD4/CD8); and (4) CD20+ B cell accumulation and altered B cell receptor (BCR) repertoire. Abbreviations: ACC: anterior cingulate cortex; CSF: cerebrospinal fluid; DPFC: dorsal prefrontal cortex; FC: frontal cortex; HP: hippocampus; SEZ: subependymal zone. Red font indicates discrepancies among studies.

**Figure 2 cells-12-02099-f002:**
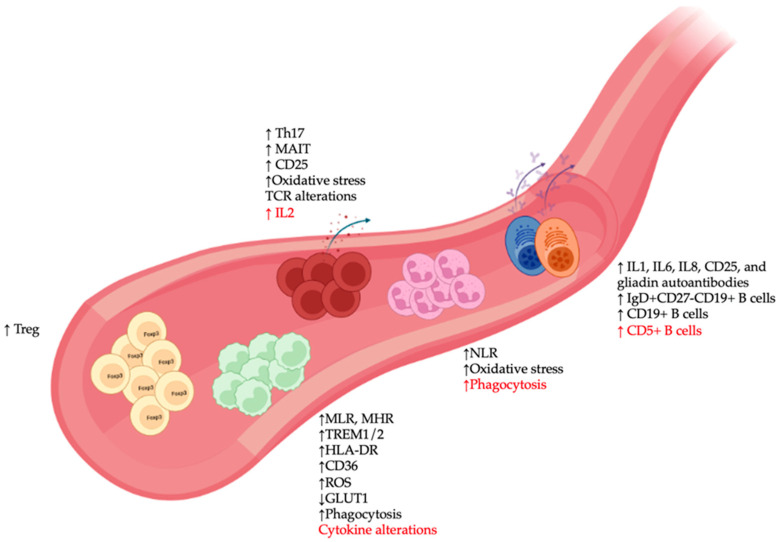
Peripheral immune alterations in patients with schizophrenia (SCZ). Major changes in immune cell types in blood samples of SCZ patients included: (1) Alterations in monocytes such as increased monocyte-to-lymphocyte and monocyte to HDL ratios (MLR and MHR), changes in expression of various immunometabolic markers (TREM1/2, HLA-DR, CD36, reactive oxygen species [ROS], GLUT1), and abnormalities in phagocytosis and cytokine production; (2) elevated expression of various neutrophil-associated markers such as neutrophil to lymphocyte ratio (NLR), oxidative stress, and phagocytosis; (3) presence of various autoantibody-producing pathogenic B cell clones, as well as increased numbers of different B cell subsets; and (4) increased activation profile of T cells (CD25), alterations in oxidative stress and T cell receptor [76] repertoire, and accumulation of immunoregulatory T lymphocyte populations, such as regulatory T (Treg), IL17-producing T-helper (Th17), and mucosal-associated invariant T (MAIT) cells. Red font indicates discrepancies among studies.

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
