# Peer review of "Microglia and Other Cellular Mediators of Immunological Dysfunction in Schizophrenia: A Narrative Synthesis of Clinical Findings"

_cells, 2023, doi:10.3390/cells12162099_

Round 1

Reviewer 1 Report

In the current manuscript, Nguyen et al. discussed the role of microglia and other cellular mediators emphasising immunological mechanisms behind schizophrenia pathophysiology.

Manuscript is well conceptualized and nicely written. Figures are also of good in graphical as scientific soundness. However, there are few minor changes need to be addressed in the manuscript, which are: 

Minor:

1. In the line 48-49, it could be...etiology of SCZ, such as exposure to toxins...."

2. Figure 1 could be placed after or in-between the subsections of the section 2.

3. For the political soundness, it is advised to use "schizophrenia patients" instead of 'schizophrenic patients' throughout the manuscript.

Major

4. It is hard to follow the long paragraph. A table should be added summarizing the role of microglia and other cellular mediators of immunological dysfunction in schizophrenia, including evidence from preclinical (animal models and in-vitro) and clinical (patients and postmortem brain) studies.

5. A section should be added summarizing developments implicating pharmacological interventions targeting neuro-immune dysfunction in SCZ.

Reviewer 2 Report

In their review article entitled "Microglia and other cellular mediators of immunological dysfunction in schizophrenia: A narrative synthesis of clinical findings",  Khoa D. Nguyen and colleagues summarize the evidence on contribution of microglia and various immune cell populations in the development of schizophrenia. The review is well written and curated and the information is organized in appropriate sections. The assocation of schizophrenia with inflammation is by now well established and the authors take extra care to highlight the studies on immune cell changes in the CNS and peripheral blood of schizophrenia patients at different stages of the disease (e.g. first episode of psychosis vs chronic or drug naive patients). I have only minor comments that may help authors to further improve clarity of presentation.

Comments

1. page 3, line 112: (IL) β  should probably be IL1β

2. page 4, line 141: It might help the reader to state what exactly TSPO is 

3. page 4, line 182: if C1qA refers to the complement component it should be stated as such for clarity

4. page 5, line 192: in this (and other parts) the authors refer to "high inflammation". It would help the reader if they can provide an accurate definition of what consittutes (i.e. what are the hallmarks) of high inflammation in the brain.

5. page 10, line 441: This is not a truly valid statement. Clozapine being the prototypical atypical APD does not act primarily on D receptors, especially D2. Perhaps the authors should rephrase.

6. page 10, line 448: Here the authors refer to T cell methylation profiles and schizophrenia. Perhaps the authors should include here a recent study showing that genetic varisnts associated with schizophrenia (and other psychiatric disorders) are enriched at epigentically active sites in lymphoid cells, especially CD4 T-cells (DOI: 10.1038/s41467-022-33885-7).

Round 2

Reviewer 1 Report

I have no further comments.